# Learning to Cycle: Is Velocity a Control Parameter for Children’s Cycle Patterns on the Balance Bike?

**DOI:** 10.3390/children9121937

**Published:** 2022-12-09

**Authors:** Cristiana Mercê, Rita Cordovil, David Catela, Flávia Galdino, Mafalda Bernardino, Mirjam Altenburg, Gonçalo António, Nancy Brígida, Marco Branco

**Affiliations:** 1Centro Interdisciplinar de Estudo da Performance Humana, CIPER, Faculdade do Motricidade Humana, Universidade de Lisboa, 1499-002 Cruz-Quebrada, Portugal; 2Departamento de Atividade Física e Saúde, Escola Superior de Desporto de Rio Maior, Instituto Politécnico de Santarém, 2040-413 Rio Maior, Portugal; 3Centro de Investigação em Qualidade de Vida, CIEQV, Instituto Politécnico de Santarém, Complexo Andaluz, 2001-904 Santarém, Portugal

**Keywords:** balance bike, children, cycle patterns, control parameter, L2Cycle program, IMU

## Abstract

The balance bike (BB) has been pointed out as being the most efficient learning bicycle due to its inherent stimulation of balance. However, the process of acquiring the control of balance on the BB has not been explored. This study aimed to: (i) categorize the cycle patterns of children on the BB, (ii) compare the cycle patterns in different stages of learning (before and after six sessions of a BB practice program), and (iii) verify whether velocity is a control parameter leading to transitions between different cycle patterns on a BB. The data were collected during the Learning to Cycle program from 12 children aged 6.06 ± 1.25 years. The velocity was measured using an inertial sensor. Seven different movement patterns were captured and categorized through video analysis. After practice, there was an increase in the mean number of different patterns and in the global mean and maximum velocity. These were interpreted as an improvement of the motor competence in the use of the BB. The results obtained support the hypothesis that velocity is a control parameter which leads to the emergence of diverse patterns of behavior. As the speed increased, the amount of foot contact with the ground became less frequent and the locomotor modes that imply that longer flight phases began to emerge.

## 1. Introduction

Riding a bicycle is an important motor milestone in a child’s life [1], given its beneficial impact on the child’s health and social development [2,3]. Consequently, investigating this learning process is essential as promoting the early onset of cycling is of considerable importance.

The dynamical system theory [4,5] provides us with an appropriate framework to study learning and development. It addresses the process of change and attempts to capture and understand the transitions that occur in complex systems [6]. The learner is considered to be a complex biological self-organized system, and movement patterns emerge out of the interactions between the different subsystems in the body, the task, and the environment [7]. According to this theory, during motor behavior or learning, the movement patterns that arise are the order parameters, or collective variables, of the system, which are constrained by the control parameters. The latter part can produce change from one movement pattern to another [5]. Due to the nonlinear behavior of the complex systems, under certain conditions, a small change in a control parameter can lead to abrupt changes in the overall system, resulting in a phase transition between the states of the system, when one state becomes a greater attractor than the previous one was. For example, an increase in treadmill velocity might not cause someone to change from walking to running, but if the velocity is increased to 2 m/s, it probably will [8], since it is not mechanically efficient to walk at that speed. Additionally, in aquatic locomotion, i.e., swimming, the change of speed influences the coordination patterns of the arms in crawl swimming [9,10]. At around the critical speed of 1.8 m/s, swimmers abruptly change from a catch-up pattern, in which the hands meet after each stroke, with only one arm moving at a time, to the relative opposition pattern, in which the propulsive phase of one arm ends when the propulsive phase of the other arm begins [9]. Even in animal locomotion, velocity is a control parameter, and this is notable in horses. Horses have three stable patterns: walking, trotting, and galloping. As the velocity increases, the horse’s locomotor system is compelled to switch from walking to trotting, and then from trotting to galloping [11]. The control parameters such as velocity can be manipulated to facilitate the emergence and transition of specific coordination patterns, which can be extremely useful for training and learning [10].

Velocity seems to be a common control parameter that moves the system through various forms of locomotion. This is true both in animals [11] and humans and in terrestrial [8] and aquatic locomotion [9,10]. It could also be a control parameter for learning how to cycle. Over the years, the process of learning to cycle has undergone changes [12]. In more recent times, a notable change has been the introduction of the balance bike (BB). A BB is a bicycle without pedals or training wheels, which is opposed to the previously standardized use of training wheels (stabilizers) on a regular bicycle [13]. The studies have indicated that the BB is a more effective learning tool than a traditional bicycle with lateral wheels is [13,14]. The lack of pedals is a task constraint that gives the children the opportunity to explore several modes of locomotion, e.g., walking, running, or gliding on the bike. These different cycle patterns can be seen as organizational stable states corresponding to different order parameters in the dynamic system theory [15]. During a glide pattern, the children do not have any direct contact with the ground since their feet are up in the air, thus, they need to explore and acquire dynamic balance with the bicycle in order to cycle [15]. Considering that balance acquisition is a key element for cycling [14,16,17], promoting gliding on the BB could enhance the children’s dynamic balance control, thereby accelerating the transition to a traditional bicycle with pedals. To better understand the learning process with a BB, it is important to categorize the different cycle patterns that might emerge while the children use it and to try to identify the control parameter that promotes the transitions between these patterns. If the control parameter can be identified, it can be manipulated in order to promote better or faster learning [10]. In this case, i.e., learning to cycle, manipulating the control parameter would promote the emergence of patterns that allow for a greater exploration of the balance component (e.g., gliding), contributing to faster learning. However, to our knowledge, no study to date has addressed this.

Considering the importance of velocity in determining the transition between the modes of locomotion in humans and animals [8,9,10,11], we hypothesized that it might also be a control parameter in the emergence of the BB’s locomotor patterns. Thus, at higher velocities, the child could be compelled to adapt to a glide pattern. Therefore, in the present study, we aimed to: (i) analyze and categorize the cycle patterns of the children on the BB, (ii) compare the patterns that emerged in different stages of learning (before and after six sessions of a BB practice program), and (iii) verify if the velocity of propulsion was a control parameter leading to transitions between the different cycle patterns exhibited on the bike.

## 2. Materials and Methods

### 2.1. Study Design

The data for this study were collected during the Learning to Cycle program (L2Cycle). This intervention program aimed to teach young children to cycle, and it included daily 30 min cycling sessions which were divided in two phases: a first phase of six sessions with BB, and a second phase of four more sessions on a traditional bicycle, i.e., with pedals. For the present study, only the data referring to the first phase of the program, i.e., with the BB, were considered. A first observation (observation 1) was conducted before the BB sessions, and a second observation (observation 2) was carried out after these sessions.

The program and the data collection were approved by the Ethics Committee of the Faculty of Human Kinetics, University of Lisbon (approval number: 22/2019), and the consent of the participants was obtained.

### 2.2. Participants

The participants in the study comprised twelve children (four girls) who were between three and seven years of age (1 child was aged 3 years, 2 children were aged 4 years, 2 children were aged 5 years, 5 children were aged 6 years and 2 children were aged 7 years, M = 6.06; SD = 1.25 years) from two kindergartens and public elementary schools in Alfragide, Portugal. None of the sample participants were able to cycle independently using a traditional bike. To be considered an independent cyclist, the child should have the ability to self-launch, ride for at least 10 consecutive meters, and brake safely.

### 2.3. Data Collection and Protocols

For each data collection moment (observations 1 and 2), each child was invited to ride a BB freely while no instructions were given for a period of five minutes on a 10 m × 10 m field.

The children’s trials on the BB were filmed using a smartphone (Samsung A71, South Korea) at 30 Hz, which was positioned in one of the field’s vertices to cover it entirely in the field of view.

The bicycle’s velocity was collected using an inertial measurement unit (IMU) (SparkFun 9DoF Razor, Niwot, CO, USA) that was secured in the spokes of the front wheel [18]. According to previous pilot testing, the IMU was sampled at 100 Hz, the accelerometer was sampled at 4 G, and the gyroscope was sampled at 2000 deg/s.

To synchronize the IMU and the video data, before the task, the researcher lifted and dropped the front wheel of the BB on the ground three times. The data were later synchronized by identifying the video frame of the first impact of the front wheel on the ground, which corresponded to the first acceleration peak in the IMU.

### 2.4. Data and Statistical Treatment

The categorization of the cycle patterns started with the first analysis of the videos to identify the potential patterns. Subsequently, the categorization criteria were discussed and elaborated by a panel of three experts; two of them were experts in child motor development, and one of them was an expert in biomechanics and movement analysis. After they came to a unanimous consensus (Table 1), the instrument inter reliability was assessed through the overall Fleiss’s kappa statistics, and the intra reliability was assessed through the overall Cohen’s kappa [19]. Thirty-five video clips were independently categorized by four independent observers (five clips for each defined pattern), revealing an overall inter reliability of k = 0.854 and an intra reliability of k = 0.921. Once a strong instrument reliability was ensured, all of the videos were visualized and categorized by the same observer using Kinovea software to identify the first and last frames of each pattern. The number of different patterns explored was collected per child and moment of observation.

The variables related to velocity were calculated with a custom matlab routine which converted the angular velocity of the front wheel into linear velocity. By synchronizing the IMU and video data, it was possible to calculate the global velocity of each child through observation, as well as the velocity of each pattern.

The frequency of the children that explored each pattern and descriptive statistics regarding global and pattern’s velocities per child were determined by moment of observation (Table 2).

The Shapiro–Wilk test was used to estimate the samples’ normality of the data distribution. Accordingly, the number of different patterns explored by each child between the two moments of observation was compared with the Wilcoxon test, and the global velocities (i.e., minimum, mean, and maximum) between the two moments of observations were compared using paired sample t-tests; the r effect size was also calculated.

Based on the data collected in observation 2, probability curves, which show the probability of each cycle pattern to occur at a given speed, were calculated (see Figure 1). A moving filter average with a span of 0.2 was applied by a method of local regression using lower weight to outliers in linear least squares and a second-degree polynomial model (rloess). The thresholds were estimated. These thresholds represent the moment when the previous pattern became less likely to occur than the next one did, indicating a phase transition in the system.

## 3. Results

### 3.1. Categorization of Cycle Patterns

Seven mutually exclusive cycle patterns were categorized: walking, running, gliding, trotting, hopping, single hopping, and jumping. The type of contact with the ground during movement (i.e., single support, double support, and flight phase) was used as the criterion to distinguish between the patterns (Table 1). For all of the patterns, excluding the glide, the start was considered to be at the first frame of the first support, and it ended at the last frame of the last support. In the glide pattern, the start was considered to be from the initial and single moment of impulse (followed by the flight phase) until the moment immediately before the subsequent contact of the foot or feet on the ground. For gliding to be considered a pattern, the child must balance on the BB without immediately searching for a new support for at least two-wheel revolutions.

### 3.2. Cycle Patterns Pre and Post Intervention

Considering all of the sample, in observation 1, four different patterns were identified: walk, run, glide, and hop, whereas in observation 2, seven patterns were identified. Between observations 1 and observation 2 a significant increase occurred in the number of different patterns performed by the children (z = −3.10, *p* = 0.002, r = −0.26), global mean velocity (t(11) = −8.50, *p* < 0.001, r = 0.93), and the global maximum velocity (t(11) = −12.89, *p* < 0.001, r = 0.97) (see Table 2).

### 3.3. Velocity as a Control Parameter

Considering all of the patterns, three of them stand out as the most frequent ones in the different velocity bands. In the lower velocities, the walking pattern prevails, achieving almost the total frequency. As the velocity increases, the walk frequency decreases, and in the velocity value of 1.32 m/s (t1), the walking and the running frequencies cross over, with the run becoming the most frequent pattern. As the velocity continues to increase, the glide frequency also increases, and it crosses the run with the value of 2.15 m/s (t2), becoming the most frequent one (see Figure 1).

## 4. Discussion

### 4.1. Categorization of Cycle Patterns

The primary objective of this study was to analyse and categorize the cycle patterns used by the children when they were riding a BB. The results revealed at least seven distinct patterns, meaning that the BB can afford a diversity of motor behaviours to the children. This possibility of achieving the same end state (i.e., riding the BB) following different paths (i.e., using distinct patterns) reflects the equifinality of the child–bicycle system [20].

So far, and despite the BB’s increasing popularity, the research specifically targeting this bicycle is still scarce. We could find two articles with suggestions for BB exercise or sessions [17,21], and only one article studied the effect of the BB sessions with preschool and/or elementary school children [22]. However, in none of them were the different patterns of locomotion analysed, defined, or categorized. Thus, the present study addresses this gap in the literature, presenting a categorization of the cycle movement patterns that emerged while the children freely used the BB on diverse surfaces, slopes, and at different velocities (L2Cycle Program). The current categorization can now be used for different purposes such as comparing the different learning paths in the use of BB, assessing the preferences according to the child’s characteristics, or monitoring each of the children’s cycling evolution.

### 4.2. Evolution of Locomotor Patterns from Pre- to Post-Intervention

Regarding the second objective, the comparison of the locomotor pattern evolution from pre- (observation 1) to post-intervention (observation 2), there are some points that should be noted. During observation 1, the children had an initial contact with the BB for five minutes, and even within such a short time frame, the children displayed four different patterns (walking, running, gliding, and hopping). These abilities emerged without any instruction, resulting solely from the children’s exploration of the constraints inherent to the child–BB system, rendering them as foundational patterns for learning. After six 30 min sessions, corresponding to three hours of potential practice, the children had explored a significantly greater number of locomotor patterns. Seven children tried all of the seven patterns, and the other five children tried six of them. The L2Cycle program was conceived based on the ecological and dynamic perspective propositions, i.e., there was a structuring of the practice environment (slopes, friction gradient, and obstacles) and a regulation of the possible control parameter (“try faster”), thus prescribing the practice conditions, but there were specific instructions as how to propel the ground (patterns). In the absence of specific instructions, the child–BB system worked as a dynamic system that was capable of self-organization in which several cycle patterns emerged as a result of the exploration of the existent constraint.

From observation 1 to 2, the children also significantly increased their global mean and maximum velocity, meaning that they not only were able to perform more patterns, but they also improved their motor efficiency on the BB. This improvement in a short period of time is in accordance with the study of Shim, Davis, Newman, Abbey and Garafalo-Peterson [22], which was a BB intervention with pre-school children who showed significant improvements in their balance after no more than three hours of practice (from 15 to 20 min sessions over the course of three weeks). In our study and in Shim et al.’s [22] study, the practice occurred throughout different days, which seems to be an advantage, because it allowed the learners to benefit from both of the motor learning modes, the on-line mode, which occurs when the learner is practicing, and also the off-line mode, in which the learner continues to acquire or stabilize the skill during sleeping or napping [23,24].

### 4.3. Velocity as a Control Parameter

The present data confirmed that the velocity can be considered a control parameter for the emergence of different locomotor patterns on the BB. Traditionally, testing the velocity as a control parameter is conducted under controlled laboratory settings using a treadmill that allows for a constant increase in the velocity, which is followed by a decrease at the same rate [25,26]. However, learning to cycle on a treadmill with changing speeds would not be a good option in terms of safety for the children, and the task would have lower ecological validity. Assessing learning in a real world context using the IMUs, small portable biomechanical devices, allowed us to capture the velocity in a reliable way [18], while the children freely explored the constraints that acted upon the child–BB system [7].

The significant increases in velocity and in the number of explored patterns between observations 1 and 2 is the first indicator that velocity is a potential control parameter. According to our data, as the exploration of the new velocity limits began, new cycle patterns emerged, which is in accordance with the definition of a control parameter that moves the system through its collective states [5].

This hypothesis is confirmed in Figure 1, which shows that there were three main preferred cycle patterns, or order parameters, on the BB: walking, running, and gliding. These results are in line with what was already known regarding the importance of velocity as a control parameter for locomotion [8] and for swimming [9,10], highlighting that also during locomotion on a bicycle, as the velocity increases, the system moves through different patterns, or action modes. For the velocities below 1.32 m/s, walking seems to a more stable action mode as it is the one that the children displayed with greater frequency. However, above 1.32 m/s (t1), running seems to become more comfortable, and that is the preferred action mode until the velocity reaches 2.15 m/s (t2), when gliding becomes the prevalent action mode. So, velocity can be considered a control parameter of the system that leads to phase transitions as the stability of the different attractors is threatened. Besides the three main cycle patterns, other action modes were explored by the children, but with a lower frequency. This multiplicity of patterns for the same velocity could represent a catastrophic multimodality flag [27]; a characteristic phenomenon of dynamical systems [5]. Those less frequent patterns reflect the children’s exploration of new solutions, which occurred mainly between 0.7 m/s and 2.5 m/s since at lower speeds, walking is clearly the strongest attractor, and at higher speeds, gliding seems to be preferred.

Interestingly, the walking and running velocities on and off the BB are similar. The mean velocity of the walking pattern on the BB was (1.12 m/s) very close to that when they were walking beside the bicycle, which is typical of developing children aged 10 (1.21 m/s) [28], and also, the maximum velocity of the running pattern on (2.48 m/s) the BB to the jogging mean velocity in those children (2.61 m/s) [28]. Similar results were also found for the 3–4 year old children for walking between 0.5 m/s and 1.5 m/s and for the 7–8 years old mean velocity between 0.5 and 2.0 m/s [29].

The present findings provide new information about the dynamics of cycling which were explored through practice with the BB. For the intermediate speeds, the children tend to explore various organizational states (cycle patterns on the bike) that afford shorter or longer flight phases with no foot contact with the ground. At the higher speeds, gliding becomes prevalent, leading the children to experience balance for longer periods, a skill which is necessary to ride a traditional bicycle. The previous literature supports the hypothesis that the BB is a better tool to learn to cycle independently when it is compared to the bicycle with lateral training wheels (BTW) because it seems to enable balance acquisition at the early stages of learning [13,14,16,17]. However, until now, it was not known which cycle patterns were more frequently explored by the children on a BB, nor what the preferred velocities were for the transition between them. Considering the importance of acquiring the gliding pattern to control balance before trying to ride a traditional bike, the children should be given the opportunity to explore different velocities while learning with the BB. This can be made easier by choosing a learning environment with small uneven ramps, or even by promoting races on the BB. Once control of the velocity and balance have been acquired, the children can move on to the traditional bike to practice pedalling and consequently learn to cycle independently with greater ease.

## 5. Conclusions

This study identified seven distinct locomotor patterns used by children while they were learning to cycle with the BB. The number of locomotor patterns explored increased as the children became more skilled on the BB (i.e., in the second observation). Walking, running, and gliding on the BB were prevalent over the other locomotor patterns, and each one was prevalent at critical values of velocity. Thus, velocity was identified as the control parameter that moves the system through its different collective states. At the higher velocities (above 2.15 m/s), gliding becomes the preferred action mode. To glide, the children need to maintain balance on the bike, which is important to facilitate the learning and acquisition of cycling on a traditional bicycle. For this reason, parents and teachers should be able to create practice conditions that potentiate learning while exploring different velocities.

## Figures and Tables

**Figure 1 children-09-01937-f001:**
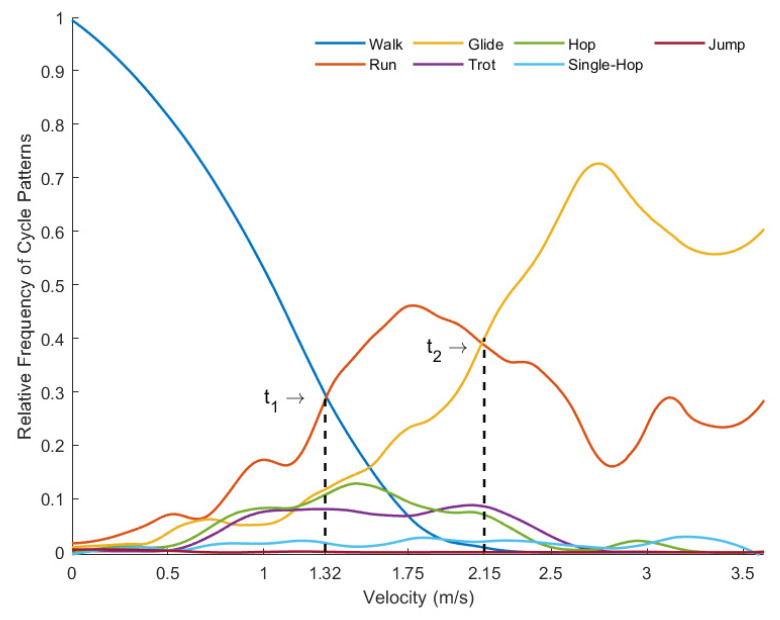
Relative frequency of each cycle pattern along the explored velocities and threshold points (t1 and t2) that indicate a shift in the prevalent cycle pattern.

**Table 1 children-09-01937-t001:** Categorization of cycle patterns on BB based on ground contact phases description.

Patterns	Description and Ground Contact Phases
Walk	Based on walking pattern. Composed of a single support phase, which is followed by a double support phase and no flight phase. Single supports are alternated 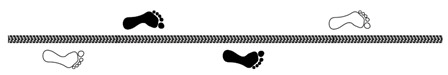
Run	Based on run pattern. Composed of single support phase, which is followed by flight phase and a new single support, single supports are alternated 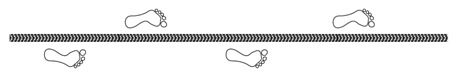
Glide	The child propels itself (through a single or double support) and maintains balance on the bicycle for at least two-wheel revolutions 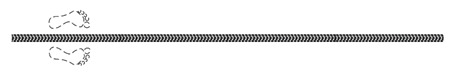
Hop	The child is propelled through a single support, which is followed by flight and a new single support on the same side. For it to be considered a hop, at least two consecutive simple supports are needed 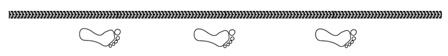
Single Hop	The child is propelled through a single support, which is followed by flight phase 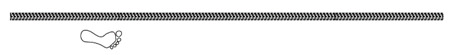
Trot	The child is propelled through a single support, which is followed by a double support phase and a flight phase 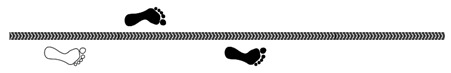
Jump	The child is propelled through a double support, which is followed by a flight phase 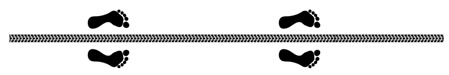

Note. 
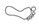
 single support; 
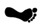
 double support; 
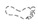
single or double support; 
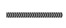
 bicycle trace.

**Table 2 children-09-01937-t002:** This is a table. Tables should be placed in the main text near to the first time they are cited.

Velocities (m/s)	Observation 1				Observation 2					
	Minimum	Mean	Maximum	N		Minimum	Mean	Maximum	N	
	M ± SD	M ± SD	M ± SD			M ± SD	M ± SD	M ± SD		
Global	0.01 ± 0.01	0.67 ± 0.26	1.31 ± 0.41	12		0.13 ± 0.30	1.60 ± 0.47	2.52 ± 0.49	12	
Walk	0.01 ± 0.01	0.68 ± 0.25	1.28 ± 0.39	12		0.17 ± 0.41	1.12 ± 0.24	1.95 ± 0.19	11	
Run	0.99 ± 0.41	1.36 ± 0.25	1.72 ± 0.03	2		0.58 ± 0.57	1.75 ± 0.44	2.48 ± 0.52	12	
Glide	1.78 ^a^	1.86 ^a^	1.90 ^a^	1		0.85 ± 0.57	1.75 ± 0.42	2.31 ± 0.60	12	
Trot				0		0.93 ± 0.71	1.68 ± 0.43	2.17 ± 0.43	12	
Hop	0.89 ± 0.03	1.11 ± 0.25	1.35 ± 0.49	2		0.93 ± 0.63	1.68 ± 0.44	2.30 ± 0.40	12	
Single Hop				0		1.04 ± 0.56	1.70 ± 0.45	2.20 ± 0.57	12	
Jump				0		1.23 ± 0.89	1.61 ± 0.70	1.91 ± 0.60	8	

Note. ^a^ SD not presented since only one episode occurred.

## Data Availability

The authors have made the data available for the journal they are required.

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
