# Peer review of "Learning to Cycle: Is Velocity a Control Parameter for Children’s Cycle Patterns on the Balance Bike?"

_children, 2022, doi:10.3390/children9121937_

Round 1

Reviewer 1 Report

Thank you for allowing me to read and review the manuscript. The study is well written and provides knowledge in child psychomotricity. I only mention some aspects that must be addressed by the authors:

L48-50. The phrase is inappropriate for the context of the study. I suggest removing the phrase or changing to walk-jog-run patterns in people. Not on horses.

L52. “Velocity can also be a control parameter for learning how to cycle”. Include reference.

L53. “Over the years, learning to cycle has seen some changes occur”. Include reference.

L59-60. “These different cycle patterns are organisational stable states that correspond to different order parameters in the dynamic system theory”. Include reference.

L60-62. “During the glide pattern, children do not have any direct contact with ground since their feet are up in the air; so, they need to explore and acquire dynamic balance with the bicycle, in order to cycle”. Include reference.

L66. “balance bike”. use abbreviation.

L71. “…and animals”. Remove animals.

L109-111. I suggest including this description in materials and methods.

L120-125. I suggest including this description in materials and methods.

Author Response

On behalf of the entire team, we would like to thank the reviewer’s comments, which we believe to have contributed for the improvement of the manuscript.

Next, we present the reviewer’s comments (at bold) and our answers below:

Thank you for allowing me to read and review the manuscript. The study is well written and provides knowledge in child psychomotricity. I only mention some aspects that must be addressed by the authors:

Response: Thank you very much for your comment, we are very pleased to hear that you liked the manuscript.

L48-50. The phrase is inappropriate for the context of the study. I suggest removing the phrase or changing to walk-jog-run patterns in people. Not on horses.

Response: Thank you for your comment. However, considering reviewer’s 2 suggestion of trying to develop the introduction section, we decided to give a few more examples of studies focused in identifying control parameters for different actions and we decided to keep the horses example. We feel that the phrase you mentioned is now less out of context, since we believe it is interesting to highlight that phase transitions between locomotion patterns are not exclusive to our species, and velocity seems to be a control parameter tranversal to humans and animals. The walk-run phase transition in humans is mentioned in lines 45-48.

L52. “Velocity can also be a control parameter for learning how to cycle”. Include reference.

Response: We have reformulated identified sentence including several references.

L53. “Over the years, learning to cycle has seen some changes occur”. Include reference.

Response: Reference added.

L59-60. “These different cycle patterns are organisational stable states that correspond to different order parameters in the dynamic system theory”. Include reference.

Response: Reference added.

L60-62. “During the glide pattern, children do not have any direct contact with ground since their feet are up in the air; so, they need to explore and acquire dynamic balance with the bicycle, in order to cycle”. Include reference.

Response: Reference added.

L66. “balance bike”. use abbreviation.

Response: Done.

L71. “…and animals”. Remove animals.

Response: Thank you for the suggestion, but since we kept the horse examples, as previously explained, we did not remove the indicated words as we choose to kept the horse study.

L109-111. I suggest including this description in materials and methods.

Response: The indicated lines are in the "Materials and Methods" section, specifically in the "Data and Statistical Treatment" sub-section, since they address the categorization instrument developed to treat the data.

L120-125. I suggest including this description in materials and methods.

Response: The indicated lines are in the "Materials and Methods" section, specifically in the "Data and Statistical Treatment" sub-section, since they address the treatment process of the variables under analysis.

Reviewer 2 Report

It is recommended to develop the Introduction section. It would be beneficial to support the problem situation with relevant research and to discuss it in more detail. It is recommended that the participants be introduced in more detail. It is recommended that the findings be supported by the relevant literature. In addition, current sources should be added to the study.

Author Response

On behalf of the entire team, we would like to thank the reviewer’s comments, which we believe to have contributed for the improvement of the manuscript.

Next, we present the reviewer’s comments (at bold) and our answers below:

It is recommended to develop the Introduction section. 

Response: Thank you for your comment, as suggested we have developed the introduction a little further, namely by adding more studies.

It would be beneficial to support the problem situation with relevant research and to discuss it in more detail.

Response: As suggested, we reinforced the literature supporting the presentation of the problem and tried to clarify it even more.

It is recommended that the participants be introduced in more detail. 

Response: Thanks for your suggestion. We have added more information about the children's school context and their age distribution.

It is recommended that the findings be supported by the relevant literature. 

Thanks for the comment. Until now, there was no study that addressed the issue of velocity as a control parameter in cycle patterns. For this reason, we do not have literature that directly supports our results. Nevertheless, we cite and compare our results with studies that also identify velocity as a control parameter in other forms of locomotion (i.e., walking-running and swimming).

In addition, current sources should be added to the study.

Response: Thank you for your comment. The addition of current literature was one of our concerns. Even though we added some references in the revised Ms., we could not find any recent study that addressed the issue of velocity as a control parameter in locomotion patterns on or off the bicycle. If you know of a recent study within this scope that want to recommend us, we will be glad to include it in the manuscript.

Round 2

Reviewer 2 Report

Corrections are appropriate